# Prevalence and Antimicrobial Resistance of *Klebsiella* Strains Isolated from a County Hospital in Romania

**DOI:** 10.3390/antibiotics10070868

**Published:** 2021-07-16

**Authors:** Alice Elena Ghenea, Ramona Cioboată, Andrei Ioan Drocaş, Eugen Nicolae Țieranu, Corina Maria Vasile, Aritina Moroşanu, Cristian George Țieranu, Alex-Ioan Salan, Mihaela Popescu, Adriana Turculeanu, Vlad Padureanu, Anca-Loredana Udriștoiu, Daniela Calina, Dan Cȃrţu, Ovidiu Mircea Zlatian

**Affiliations:** 1Department of Bacteriology-Virology-Parasitology, University of Medicine and Pharmacy of Craiova, 200349 Craiova, Romania; gaman_alice@yahoo.com (A.E.G.); adriana_turculeanu@yahoo.com (A.T.); ovidiu.zlatian@gmail.com (O.M.Z.); 2Department of Pneumology, University of Medicine and Pharmacy of Craiova, 200349 Craiova, Romania; ramona.cioboata@umfcv.ro; 3Department of Urology, University of Medicine and Pharmacy of Craiova, 200349 Craiova, Romania; andrei_drocas@yahoo.com; 4Department of Cardiology, University of Medicine and Pharmacy of Craiova, 200349 Craiova, Romania; tieranueugen@gmail.com; 5Department of Pediatric Cardiology, County Clinical Emergency Hospital of Craiova, 200349 Craiova, Romania; 6Department of Paediatrics, University of Medicine and Pharmacy of Craiova, 200349 Craiova, Romania; inamorosanu@yahoo.com; 7Department of Gastroenterology, Carol Davila University of Medicine and Pharmacy, 050474 Bucharest, Romania; tieranucristian@gmail.com; 8Department of Oral and Maxillofacial Surgery, University of Medicine and Pharmacy Craiova, 200349 Craiova, Romania; alex.salan@umfcv.ro; 9Department of Endocrinology, University of Medicine and Pharmacy of Craiova, 200349 Craiova, Romania; 10Department of Internal Medicine, University of Medicine and Pharmacy of Craiova, 200349 Craiova, Romania; vldpadureanu@yahoo.com; 11Faculty of Automation, Computers and Electronics, University of Craiova, 200776 Craiova, Romania; anca.udristoiu@edu.ucv.ro; 12Department of Clinical Pharmacy, University of Medicine and Pharmacy of Craiova, 200349 Craiova, Romania; calinadaniela@gmail.com; 131st Department of Surgery, University of Medicine and Pharmacy of Craiova, 200349 Craiova, Romania; cartu_dan@hotmail.com

**Keywords:** multidrug-resistant bacteria, *Klebsiella*, antibiotic resistance

## Abstract

The study evaluated the evolution of the incidence of infections with *Klebsiella* in the County Clinical Emergency Hospital of Craiova (SCJUC), Romania. Also, we monitored antibiotic resistance over more than two years and detected changes in resistance to various antimicrobial agents. Our study included 2062 patients (823 women and 1239 men) hospitalised in SCJUC during the period 1st of September 2017 to 30 June 2019. In 458 patients (22.21%) from the 2062 total patients, the collected samples (1116) were positive and from those, we isolated 251 strains of *Klebsiella* spp. We conducted a longitudinal analysis of the prevalence of *Klebsiella* spp. over calendar months, which showed a prevalence in surgical wards that ranged between 5.25% and 19.49% in June 2018, while in medical wards the variation was much wider, between 5.15% and 17.36% in April 2018. *Klebsiella* spp. strains showed significant resistance to Amoxicillin/Clavulanate, Aztreonam and Cephalosporins such as Ceftriaxone, Ceftazidime and Cefepime. We examined the possible link with the consumption of antibiotics in the same month by performing a multiple linear regression analysis. The evolution of antibiotic resistance in *Klebsiella* was correlated with the variation of resistance in other bacteria, which suggests common resistance mechanisms in the hospital environment. By performing the regression for dependency between antibiotic resistance and antibiotic consumption, we observed some correlations between antibiotic consumption and the development of antibiotic resistance after 1, 2 and even 3 months (e.g., resistance to meropenem was influenced by the consumption in the hospital ward of imipenem 1 month and two months before, but only 1 month before by the consumption of meropenem). The clustering of strains showed filiation between multiresistant *Klebsiella* spp. strains isolated from specific patients from the ICU. The evolution of prevalence and antibiotic resistance in *Klebsiella* correlated with the resistance in other bacteria, which suggest common resistance mechanisms in the hospital environment, and also with the consumption of antibiotics.

## 1. Introduction

The primary proposed mechanism underlying antibiotic disease associations is microbiota dysbiosis, which results in changes in gene expression, epigenetic alteration, and invasion by pathogenic bacteria, the development of biofilms, and immune regulation and inflammation. This connection between antibiotic exposure and disease risk contributes to an increasing amount of knowledge that antibiotic use has harmful long-term health consequences. Significant studies have found a correlation between antibiotic use and chronic disease in both paediatric and adult populations [1].

Nosocomial infections, due to their frequency and severity, are one of the major epidemiological problems of any inpatient, regardless of the number of beds or the profile of the ward [2]. In the USA alone, there were more than 1.7 million nosocomial infections, which resulted in the near 100,000 deaths [3]. Knowledge of the epidemiological process in these infections, with all particularities that the factors of this process may present depending on the etiological agent, clinical manifestations, type of hospital unit, largely conditions the effectiveness of practical actions of prophylaxis and control. The major factors causing the development of bacterial resistance include the indiscriminate use of antimicrobial agents in human and animal medicine, agriculture, and aquatic farming [1,4]. Different mechanisms or their combination are used by bacteria in developing resistance to antibiotics [5]. The presence of plasmids that contain one or more resistance genes, with each encoding a single antibiotic resistance (AR) phenotype, often causes the development of multiple AR (MAR) in bacteria [6]. These AR genes can transfer to other bacteria of the same or different species.

*Klebsiella pneumoniae* (*K. pneumoniae*) is the second most common gram-negative pathogen after *Escherichia coli*
*(E. coli)* associated with a wide range of infections, such as urinary tract infection (UTI), pneumonia and pleurisy, intra-abdominal infections, bloodstream infections, meningitis, and pyogenic liver abscesses [7,8,9,10,11], especially in immunodeficient patients [12].

*Klebsiella* spp. are widespread in nature: in the atmosphere, where they arise from surface water, waste, soil, and plants [13,14], and the mucosal surfaces of mammals such as humans, horses, or swine, which they colonize. *Klebsiella* is similar to *Enterobacter* and *Citrobacter* in this sense, but not to *Shigella* spp. or *E. coli*, which are popular in humans but not in the ecosystem. *K. pneumoniae* is identified as a saprophyte in the nasopharynx and the gastrointestinal tract in humans. Carrier rates differ tremendously between studies [15,16].

In recent decades, rates of broad-spectrum cephalosporin-resistant *K. pneumoniae* that produce broad-spectrum β-lactamases (ESBL) have increased dramatically worldwide, and in most parts of the world, *K. pneumoniae* is the pathogen most commonly associated with the dissemination of ESBL and other horizontally transmissible resistance genes [17,18,19].

For *K. pneumoniae*, a high frequency of resistance to third-generation cephalosporins, fluoroquinolones, and aminoglycosides, has become evident in southern, central and Eastern Europe [18,20]. Many of these strains have gained resistance to all classes of antibiotics. Carbapenem resistance in *K. pneumoniae* isolates is on the rise in the European Union, except for Greece, where it is already established [20,21,22]. This is a particularly worrying phenomenon, as carbapenems are the ultimate antibiotics and treatment options for patients infected with this bacterium and other carbapenem-resistant bacteria are severely limited [23].

Because of the development of MAR strains, several *Klebsiella* sp. infections acquired in clinics raise a medical challenge. It was first reported in 1981 that *Klebsiella* strains were resistant to different generations of cephalosporins, especially the third generation; since then, these bacteria have become more resistant to antibiotics [23,24].

This study analyzed the evolution of the prevalence of infections with *Klebsiella* in the Intensive Care Unit (ICU), medical and surgical services of the County Clinical Emergency Hospital of Craiova (SCJUC), Romania, one of the largest hospitals in our country. We also wanted to determine the changes in the antibiotic resistance spectrum for two years and to observe correlations between resistances to different antibiotics over time. Also, we wanted to investigate a possible explanation for antibiotic resistance, namely the consumption of the antibiotic, which will help to adjust the hospital specific anti-biotherapy guidelines to minimize the development of resistant strains in our hospital. We also wanted to determine the degree of similarity of the strains based on antibiotic resistance profiles in an effort of epidemiologic tracing of the circulation of *Klebsiella* strains in our hospital.

## 2. Materials and Methods

For our study, we collected 2456 samples from 2062 patients (823 women and 1239 men) hospitalised in SCJUC during the period 1st of September 2017 to 30 June 2019, in the Intensive Care Unit (ICU), medical and surgical wards. The biological samples were joint fluid, bile, blood, catheter, pleural fluid, cerebrospinal fluid, purulent secretion, sputum, urine, tracheal aspirate, pharyngeal swab, nasal swab, puncture liquid, conjunctival secretion and ear discharges We first identified bacteria by classical microbiological methods, but then we verified the identification of *Klebsiella* with the automated analyzer Vitek 2 (Biomerieux, Marcy-l’Étoile, France). For most strains, antimicrobial susceptibility testing was performed by the disk diffusion Kirby-Bauer method, using the following antibiotics: Sulphametoxazole/Thrimetoprim (1.25/23.75 µg), Tigecyclin (15 µg), Ciprofloxacin (5 µg), Amikacin (30 µg), Meropenem (10 µg), Ceftriaxone (30 µg), Ceftazidime (30 µg), Cefepime (30 µg), Cefazolin (30 µg), Piperacillin/Tazobactam (100/10 µg) and Amoxicillin/Clavulanate (20/10 µg). The multiresistant strains were also tested on the automated analyzer Vitek 2 (which uses the microdilution method) and can provide Minimum Inhibitory Concentration (MIC) as a quantitative measure of antibiotic resistance.

### Statistical Analysis

The patients, specimens collected, and isolated strains, along with antibiotic resistance profiles were registered in the WHONET bacteriological analysis software, provided by World Health Organization (WHO).

Resistance phenotypes were generated for each isolate and statistically processed by the specialised module of the WHONET program. A single phenotype was considered for samples collected from the same patient if they were collected at a distance of fewer than 7 days. The SatScan module with the space-time permutation method was used to define transmission foci.

The database was then transferred to STATA. Continuous variables are presented as means ± standard deviation, and discrete variables as numbers and/or percentages. Most of the statistical analyses were performed using the STATA program (STATACORP, College Station, TX, USA, 2015). The monthly prevalences were calculated as the number of patients from which the species of interest was identified divided by the total number of patients analyzed in the respective month.

The MAR index is a reliable, accurate, and premium tool for documenting the origins of antibiotic-resistant bacteria. The MAR index is measured as the ratio of the number of antibiotics to which an organism is resistant and the total number of antibiotics to which the entity has been exposed. A MAR greater than 0.2 indicates that the high-risk source of infection is a region where drugs are widely used [25]. We used the MAR index to generate histograms for the distribution of the index in various wards to identify resistant and susceptible populations. Further, we performed one-way ANOVA on the MAR using the type of ward (ICU, surgical or medical) as a categorical variable to compare resistance levels in various types of wards.

To further explain the time variation of resistance in *Klebsiella* strains we correlated antibiotic resistance with antibiotic consumption in the same month and ward. The consumption of antibiotics was calculated in Defined Daily Doses (DDD) per 100 patient-days as follows: the consumption in tablets or vials of antibiotics ordered for a particular patient was obtained from the pharmacy, as well as the duration of treatment. We summed the doses administered obtaining the total quantity of antibiotic, that we then divided by the number of days and multiplied by 100 (to obtain DDD’s for 100 patient-days). For all patients in the ward, we averaged the DDDs.

The monthly resistances were calculated by averaging the MAR index of the *Klebsiella* strains analyzed in the month. The trending evolution was performed by testing the trend through the Chi2 test for trend.

We performed a time series analysis using the lagging antibiotic consumption as a predictor for the resistance.

The lineage of the strains was analyzed by the hierarchical clustering method in STATA software based on the MAR resistance index.

## 3. Results

After performing microbiological diagnosis by the classical and automated method (Vitek 2 analyzer for bacterial identification and antibiogram), we isolated 1503 bacterial strains from 1116 positive samples (45.44%) collected from 458 patients (22.21% from all analysed patients). The isolated bacteria included *E. coli*, *Klebsiella* spp., non-fermenter Gram-negative rods (NFR) (*Pseudomonas aeruginosa*, *Acinetobacter* spp.), *Staphylococcus aureus, Streptococcus* spp. and other species.

### 3.1. Prevalence of Klebsiella spp. Strains

In all samples, the prevalence of *Klebsiella* spp. was 10.22%, respectively 251 strains.

The prevalence of *Klebsiella* spp. had a relatively uniform distribution in all wards analysed (Figure 1), which suggests ubiquity in SCJUC. The maximum prevalence was recorded in the departments of ICU (26.67%), Thoracic surgery (23.22%), Urology (22.58%), medical wards (21.32%) and General surgery (20.98%), where it is a causative agent of decubitus pneumonia. The lowest prevalences were recorded in the Neurosurgery (8.25%), Endocrinology (7.00%) and oral maxillofacial surgery (5.00%) departments.

The *Klebsiella* strains were isolated with the highest rate from ear discharges (42.86%), even only 7 such specimens were analysed, followed by sputum (27.34%) and tracheal aspirate (21.03%). The prevalence was between 10% and 20% in blood, purulent secretion, nasal swab, and conjunctival secretion (Table 1).

### 3.2. Evolution of the Prevalence of Klebsiella Strains

We conducted a longitudinal analysis of the prevalence of *Klebsiella* spp. over calendar months, which showed a prevalence in surgical wards that ranged between 5.25% and 19.49% in June 2018, while in medical wards, the variation ranged between 5.15% and 17.36% in April 2018 (Figure 2a,b).

### 3.3. Resistance to Antibiotics of the Isolated Bacterial Species

By performing the analysis of the antibiotic susceptibility spectrum (Figure 3), we found that the highest resistance index was found in the ICU ward (58.20%), followed by surgical departments (44.69%) and the medical wards with an index of 47.32%. Overall, the mean index was 50.75%, with a significant variation depending on the type of section (one-way ANOVA test, *p* < 0.001). The post-hoc Dunnet test identified a significantly lower resistance in surgical wards (−16.41%) and medical wards (−12.37%).

By analysing the distribution of *Klebsiella* spp. resistance in the SCJUC wards, a population of multidrug-resistant strains with MAR > 0.8 was present in General Surgery, Plastic Surgery, and Pediatric Surgery wards. We observed two populations of *Klebsiella* spp. into the ICU, one with a low MAR, and another with a high MAR, corresponding to community and hospital strains.

*Klebsiella* strains showed significant resistance to cephalosporins due to hospital circulation of ESBL strains in the hospital. Thus, we observed increased resistance of *Klebsiella* to Amoxicillin/Clavulanate (85.31%), Cefazoline (78.90%), Ceftazidime (58.89%), Ceftriaxone (53.51%) and Cefepime (43.50%). Note the resistance to carbapenems due to the circulation of plasmids encoding carbapenemase-producing genes. Thus, the resistance to Meropenem in *Klebsiella* was 27.78%. Resistance to quinolones (Ciprofloxacin) in *Klebsiella* strains was 30.57% (Figure 4).

### 3.4. Analyse the Variation in Time of the Antibiotic Resistance

We used Ceftriaxone resistance as a marker for beta-lactams resistance. The longitudinal analysis of the monthly resistance index is presented in Figure 5. The resistance to 3rd generation cephalosporins oscillated between 60% and 90% in an undulant manner. We observed that the Tigecycline resistance began to increase in our hospital.

One can easily observe some parallelisms between evolutions of resistances (this parallel variation is analyzed by the time series procedures further in the text). For example, we can observe that between April and October 2018 a parallel evolution of resistance to cephalosporins and Ciprofloxacin. In early 2019, the resistance to all antibiotics remained mainly unchanged.

Next, we compared the evolution of resistance to various classes of antibiotics between *Klebsiella* strains and other Gram-negative bacilli (*E. coli* and non-fermenters) and Gram-positive cocci (*Staphylococcus aureus*), to identify correlations that can be explained by consumption-driven resistance and shared resistance genes.

The resistance of *Klebsiella* to quinolones represented by Ciprofloxacin correlated significantly with that of *Staphylococcus aureus* (r = 0.4850, *p* = 0.0302) (Table 2).

As for the resistance of Gram-negative rods to carbapenems, represented by Ertapenem, that was slightly ascending in the last year for the non-fermenters and stationary for *Klebsiella*. From Figure 6, we can also observe the synchronous evolution of carbapenem resistance rates in *Klebsiella* and *E. coli*, although the correlation coefficient was non-significant, suggesting both consumption-driven resistance and shared resistance genes. Interestingly, it can also be seen from Figure 6 that the increase of resistance in *E. col**i* seems to precede that of *Klebsiella*.

Taking into account the monthly resistance index, *Klebsiella* resistance correlated with *E. coli* (r = 0.5107, *p* < 0.0214), as *Klebsiella* is a known plasmid collector of resistance from other enterobacteria, and also we observed a negative correlation between *Pseudomonas aeruginosa* and *Staphylococcus aureus* (r = −0.4089, *p* = 0.0422) (Table 3).

To understand the sources of variation in antibiotic resistance of *Klebsiella* strains from our hospital we examined the possible link of resistance index with consumption of antibiotics (Table 4) in the same month by performing a multiple linear regression analysis (Table 5) using *Klebsiella* MAR as the outcome variable and the consumption of various antibiotics (expressed in DDDs administered in the same ward over the current month) as predictors. We chose the option to display the standardized coefficients to be able to compare the relative influence of antibiotics on resistance.

The antibiotic consumption showed marked differences between the wards, as in almost any speciality it is possible to find patients with severe infections that require high-dose antibiotherapy. We observed that Ceftriaxone is a popular antibiotic being used at dosages up to ten times than those recommended by the WHO in the wards of ICU, Neurosurgery, and other surgical wards. Also, Piperacillin/Tazobactam was used in double quantities in ICU, as Amoxicillin/Clavulanate in Cardiology.

We observed a significant effect on the antibiotic resistance index of consumption of Tigecycline (1.18%), Piperacillin/Tazobactam (−0.28%), and Meropenem (0.32%). The constant term signifies the resistance index predicted by the model that is not dependent on antibiotic consumption (intrinsic resistance).

The results of the time series analysis of the effect of antibiotic consumption on antibiotic resistance are presented in Table 6.

We observed a marginally significant correlation between the resistance to Meropenem and the consumption of Imipenem 1 month before (*p* = 0.078), but surprisingly not with the consumption of Meropenem. Nevertheless, the resistance to Meropenem strongly correlated with the consumption of imipenem 2 months before and also with the consumption of Meropenem.

We also observed that the resistance to Ciprofloxacin was strongly correlated with the consumption of Imipenem 2 and 3 months before (*p* < 0.001 for both correlations), but only with the consumption of Ciprofloxacin 2 months before (*p* = 0.005) and not 3 months before.

### 3.5. Analysis of Strains Relatedness Based on Clustering on MAR Index

*Klebsiella* presented two main clusters corresponding to the community and hospital strains (which had low resistance (blue) and high resistance (red)) (Figure 7), each with two secondary clusters of low/high resistance. Most surgical strains had a medium resistance, but the high-resistance strains were found in the Thoracic Surgery ward, Orthopaedics and ICU. It can be observed from Figure 7 that the ICU strains are divided into 3 subclusters each with 2 strains. 3 strains from Orthopaedics belong in the same cluster and Oncology strains were divided into 2 subclusters each with 2 strains.

### 3.6. Epidemiologic Tracing of the Antibiotic Resistance Phenotypes

We used antibiotics for which the resistance was common on *Klebsiella*: Aztreonam (ATM) Ceftazidime (CAZ), Cefepime (FEP) Amoxicillin/Clavulanate (AMC) TPZ: Piperacillin/Tazobactam.), Ertapenem (ETP), Ciprofloxacin (CIP) Imipenem (IMP), Amikacin (AMK) to search for repeated resistance profiles. Next, we analyzed the frequencies of each resistance profile to the above-mentioned antibiotics, as multiple strains with the same resistance profile are probably related and can represent an epidemiologic transmission event.

In this regard, we observed that the *Klebsiella* phenotype with ATM-CAZ-FEP-ACM-TPZ-ETP-CIP resistance profile appeared sporadically every two weeks, from week 32 of 2018 to week 36 of the same year, to reappear at week 41 (Figure 8), suggesting a possible resistance transmission chain.

## 4. Discussion

Chronically underreported, present anywhere in the world, nosocomial infections (IN) are and continue to be a challenge in inpatient or outpatient medicine, even in this century, due to weak spots of hygiene in hospitals, negligence of the medical staff, outdated antibiotic therapies, and other reasons. These infections are one of the leading causes of death in hospitals worldwide [2,3].

We assist in recent years with an increase in the number of *K. pneumoniae* isolates, especially antibiotic-resistant strains, particularly in high-risk wards as ICUs. The dissemination of hospital strains into the community can lead to problems of public health that should not be taken lightly [18,24,26]. Particularly in ICUs, often more than half of the isolated strains are antibiotic-resistant [27,28]. These resistant strains complicate the management of infections and particular clones spread in the hospital environment. In our hospital, *Klebsiella* strains were 26.67% of the strains isolated from the ICU department, followed by a prevalence of 23.22% in Thoracic Surgery and Urology (22.58%). We observed pretty high prevalences of *Klebsiella* in many wards, which suggests the ubiquity of *Klebsiella* in our hospital.

The transmission of *Klebsiella* strains through the air is supported by the high prevalence in specimens collected from the respiratory tract: sputum (27.34%), tracheal aspirate (21.03%) and nasal swabs (13.16%). Other studies incriminated also hospital air as a route of transmission for *Klebsiella* strains [29].

### 4.1. Prevalence of Klebsiella

Our study showed an undulant evolution of the *Klebsiella* prevalence in our hospital, without an ascending or descending trend (Chi2 test for trend *p* = 0.654). This can be regarded as a positive result because numerous studies show an increase in prevalence rates of *K. pneumoniae*, while other studies show a decrease in prevalence after applying infection control measures [30,31]. In our hospital, infection control was greatly improved in recent years. Still, the lack of a descending trend in *Klebsiella* prevalence can suggest an actual increase in incidence that was kept under control by the increase of preventive measures. The lack of seasonal variability of the prevalence of *Klebsiella* infections in our hospital can also suggest that a great percentage of these infections is actually iatrogenic in nature.

### 4.2. Antibiotic Resistance of Klebsiella

By analysis of MAR, we detected that the strains isolated from the ICU had the highest resistance, presumably due to the higher dosage of antibiotics used on these patients which often has severe infections due to lower immunity. We can also suppose in the ICU there is the transmission of multiresistant strains established as hospital multidrug-resistant strains. The histogram analysis showed in the ICU a low-resistance *Klebsiella* population with a low MAR, which also had MIC determined by the Vitek2 system to various antibiotics below epidemiological breakpoints (wild-type strains without detectable resistance mechanisms), representing about half of the *Klebsiella* strains isolated from this ward. These community strains were present in most wards, coming from recently hospitalised patients who did not fall within the definition of nosocomial infection (infection occurring more than 2 days after the date of hospitalisation). There are two more populations of *Klebsiella* in ICU, one with moderate resistance and one with increased resistance, corresponding to hospital strains.

All over the globe, the prevalence of *Klebsiella* strains and especially of multiresistant ones is increased in settings in which patients are often under antibiotic treatment. These include the ICUs where there are patients with septic shock that are massively treated with antibiotics and some surgical wards in which wounds sometimes get infected and need prolonged antibiotic treatment with doses higher than usual [27]. Our study showed the presence of resistant *Klebsiella* populations in General surgery, Plastic surgery, Pediatric surgery and ICU wards.

The multiresistant *Klebsiella* strains isolated from hospitals possess various resistance mechanisms as ESBLs, AmpC (that give resistance to penicillin and first-generation cephalosporins) [27], or carbapenemases (that confer resistance to carbapenems and 3rd generation cephalosporins) [32] that reflect in the antibiotic resistance pattern. In our study, we also observed multiple antibiotic resistances in *Klebsiella* strains, particularly high resistances were encountered to Amoxicillin/Clavulanate, 1st generation cephalosporins and to a lesser extent to the third-generation cephalosporins, which is due to the transmission of plasmids encoding resistance genes, in particular ESBL [33,34,35].

In hospitals, there is a high probability of horizontal gene transfers of resistance genes that can explain the increased resistance to Amoxicillin/Clavulanate, Ceftazidime and Ceftriaxone observed by us. Previous studies have shown an increased prevalence of OXA-48 carbapenemases in *Klebsiella* strains from SCJUC [8,36].

Due to the presence of beta-lactam resistance genes, the resistance to 3rd generation cephalosporins oscillated between 60% and 90% in an undulant manner that can be related to the periodic changes in antibiotic pre-surgery prophylaxis protocols in surgical wards, as part of infection control efforts in our hospital, which are probably the explanation to a marked decrease in resistance to Meropenem observed in the second year of the study.

The explanation of correlations between antibiotic resistances first resides in the usage of the same antibiotics in different wards, according to infection control protocols. Ciprofloxacin resistance correlation between *Klebsiella* and *Staphylococcus* is explained by the rapid development of AR to quinolones in patients with mixed infections [37]. Also, the correlation between the resistance of *E. coli* rods and *Klebsiella* is that carbapenem treatment induces resistance in both bacteria, alongside horizontal gene transfer of carbapenemases’ genes which were captured by the *Enterobacteriaceae* from *Pseudomonas* and other non-fermenter Gram-negative rods (NFR) [27,38,39]. In our results, the increase of resistance in *E. coli* seems to precede that of *Klebsiella* spp. (Figure 6), suggesting that the later acquired resistance genes from *E. coli*, as *Klebsiella* is a well-known collector of resistance genes [34,40].

By analysing the evolution in time of the resistance index of *Klebsiella* spp. strains, we observed that the graph resembles the evolution for *Pseudomonas* strains because both Gram-negative bacteria share common resistance genes. The resistance genes can be horizontally transmitted between hospital strains [18,34,35]. Thus, the virulent strain may persist for long periods and disseminate in clinical settings, because they survive to usual antibiotic treatments applied to patients according to various guidelines.

### 4.3. The Link between the Antibiotic Resistance of Klebsiella and the Consumption of Antibiotics

One of the well-known causes of the continuously increasing antibiotic resistance of bacteria worldwide is the over-prescription and consumption of antibiotics. We investigated this possible cause by performing two separated regression analyses: a multiple regression which examined the link between resistance and consumption of antibiotics in the same month and a time-series analysis which examined the link between resistance and consumption of antibiotics in the previous months, accounting for relationships between different points in time within a single series. That type of analysis should be more sensitive than the simple regression which does not account for the temporal nature of data.

By performing the regression for dependency between antibiotic resistance and antibiotic consumption, we observed, interestingly that entering the consumption of Ampicillin and Teicoplanin in the regression equation renders their regression coefficients significant, i.e., the correlation is now statistically significant. That is because of collinearity (correlation between two explanatory variables, used to explain the output variable). Indeed, both antibiotics are used in treating Gram-positive infections and their consumption is therefore correlated. Also, the consumption of Ampicillin correlates with the consumption of Meropenem, introducing further collinearity into the equation. Meropenem consumption is also correlated with imipenem consumption because the 2 antibiotics are of the same class and can be replaced with each other. For these reasons, we decided to drop Ampicillin and Teicoplanin resistances from the multiple regression models.

We observed some correlations between antibiotic consumption and the development of antibiotic resistance after 1, 2 and even 3 months. For example, resistance to Meropenem was influenced by the consumption in the hospital ward of imipenem 1 month and two months before, but only 1 month before for the consumption of Meropenem, suggesting that imipenem is a stronger and for a longer time inducer of carbapenem resistance compared with Meropenem. Indeed, in vitro studies showed that subinhibitory concentrations of imipenem can induce resistance to beta-lactam antibiotics by inducing the beta-lactamase production because its chemical structure is more related to the original penem nucleus than Meropenem [27].

Phenotypic clustering of bacterial strain relatedness can substitute genetic-based clustering for epidemiological tracing for infection control purposes [32]. We successfully realized a phenotypical clustering in the premiere for our hospital, which will have found a basis for future infection control efforts. As expected, the clustering showed filiation e.g., between multiresistant *Klebsiella* strains isolated from specific patients from the ICU, which can greatly improve the effectiveness of infection control measures. So the clustering based on MAR can at least orient the epidemiologist to the transmission events, although it cannot substitute to molecular methods.

The epidemiologic filiation tree of strains circulating within the hospital can be constructed by monitoring and linking in clusters specific resistance phenotypes. We managed to do that using the WHONET software capabilities. Thus, the resistance data from routine antimicrobial susceptibility testing performed in each clinical microbiology laborawe put limitations as a title of this sectiontory is a major resource for resistance surveillance that can be used to contribute to surveillance networks [41].

The time-series analysis can also be used for forecasting. Because antibiotic resistance can be induced by antibiotic consumption in previous months, it is possible to forecast the resistance for one or two months based on the antibiotic consumption in a certain month.

Antibiotic resistance testing can be influenced by a variety of laboratory or clinical factors. Although we analyzed all specimens collected, there is the possibility that bacteria did not grow from multiple causes like the quality of laboratory media or treatments with antibiotics. A factor that can influence the results of the study is the presence of uncultured bacteria which may be reluctant to be grown under laboratory conditions.

## 5. Conclusions

This work showed the importance of continuous monitoring of antibiotic resistance in hospitals to identify epidemiological transmission chains, adjust the antibiotherapy in various wards and improving the cure rate of infections treated in the hospital.

This can be very useful for infections that can change the guidelines of antibiotic prescription in a healthcare setting.

## Figures and Tables

**Figure 1 antibiotics-10-00868-f001:**
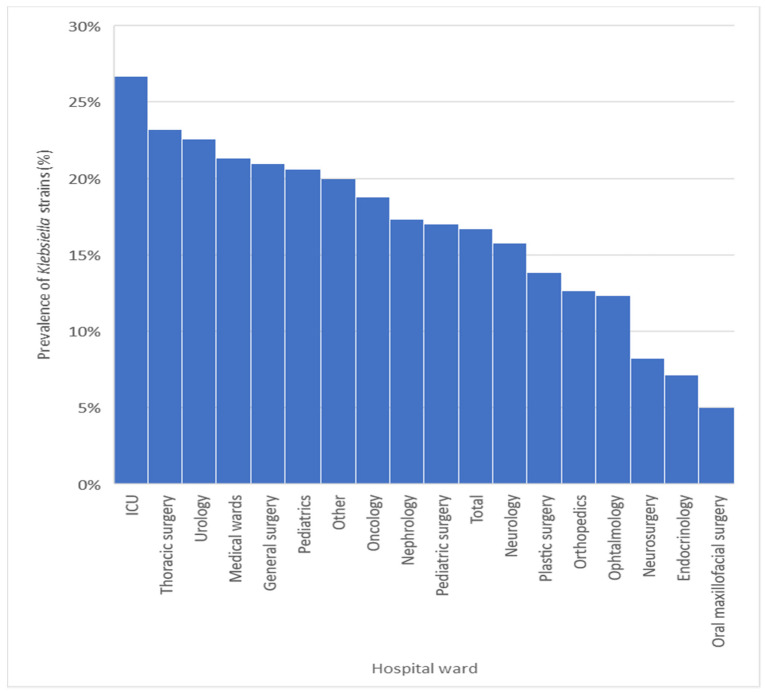
Prevalence of *Klebsiella* strains by hospital ward. The bars heights are the percentages of prevalence in different wards.

**Figure 2 antibiotics-10-00868-f002:**
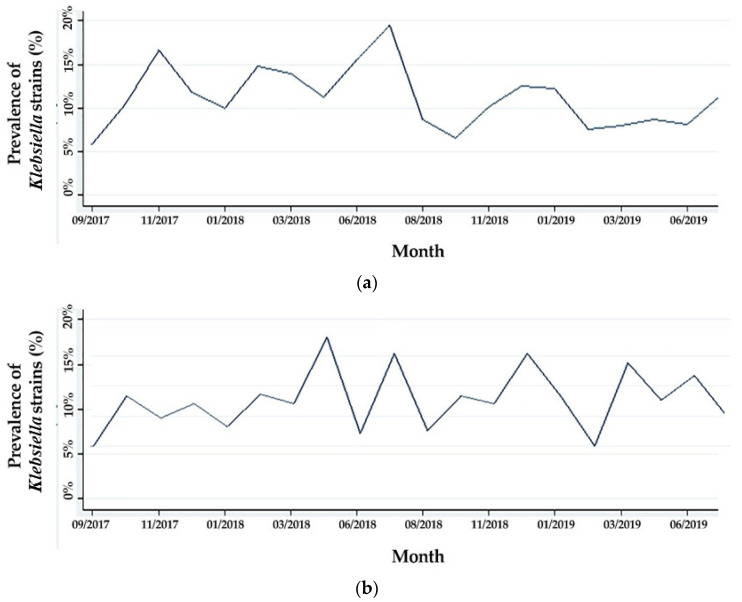
Evolution of the incidence of *Klebsiella* strains: (**a**) in the surgical wards; (**b**) in medical wards.

**Figure 3 antibiotics-10-00868-f003:**
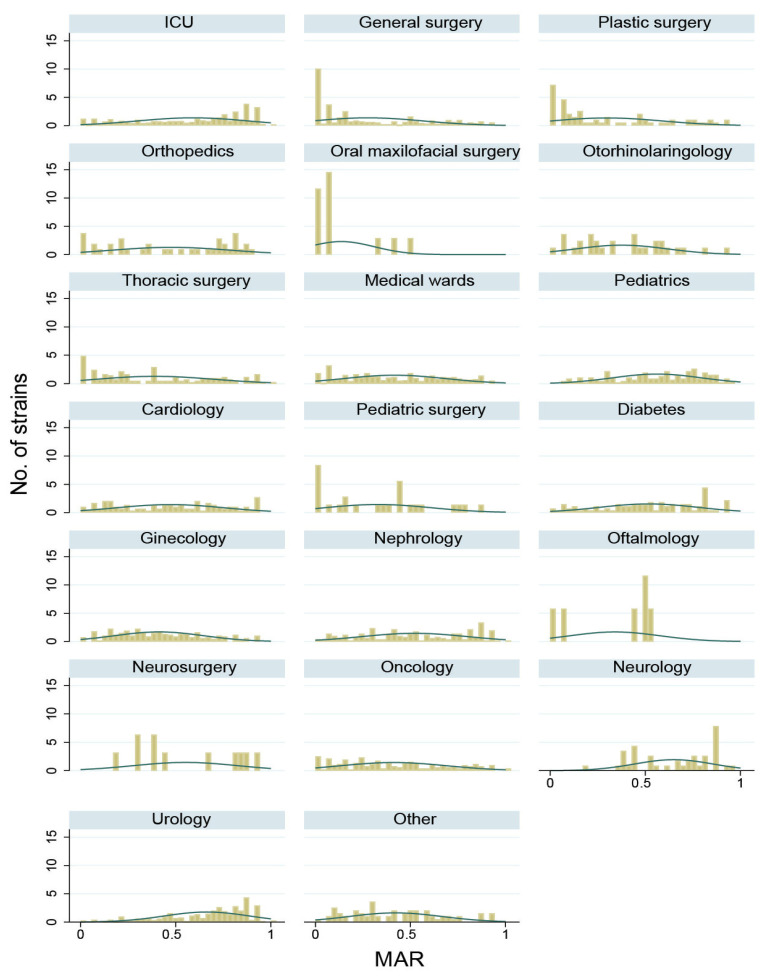
Histogram of the distribution of MAR index in various wards from SCJUC. The columns heights are the number of strains for which MAR falls in a specific interval (bin). The green line is a smoothing line of the heights of the bars.

**Figure 4 antibiotics-10-00868-f004:**
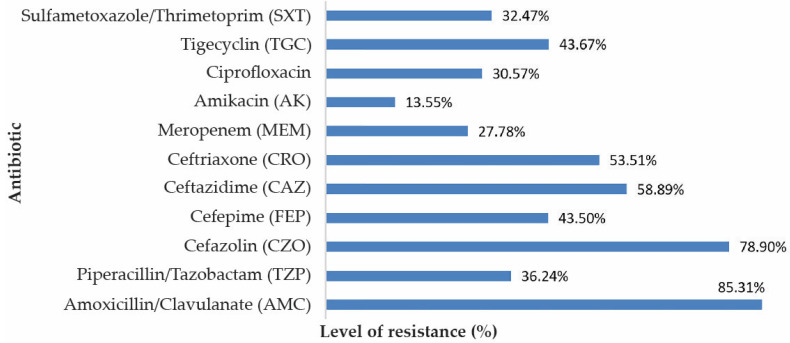
Resistance to antimicrobial substances of the isolated *Klebsiella* strains.

**Figure 5 antibiotics-10-00868-f005:**
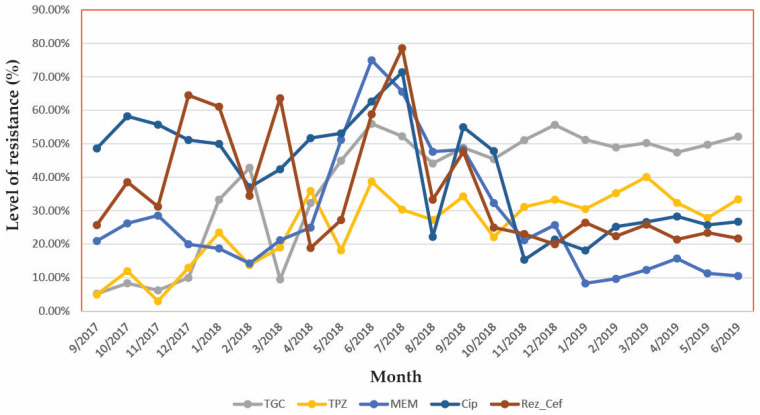
Evolution of antibiotic resistance in *Klebsiella* strains. MAR = Multiple Antibiotic Resistance index. Note: here were presented the resistance for typical antibiotics from each chemical class, however, because multiple cephalosporins were used we calculated MAR index specific for cephalosporins. TGC: Tigecyclin; TPZ: Piperacillin/Tazobactam; MEM: Meropenem; CIP: Ciprofloxacin; Rez_Cef: Resistance index to cephalosporins.

**Figure 6 antibiotics-10-00868-f006:**
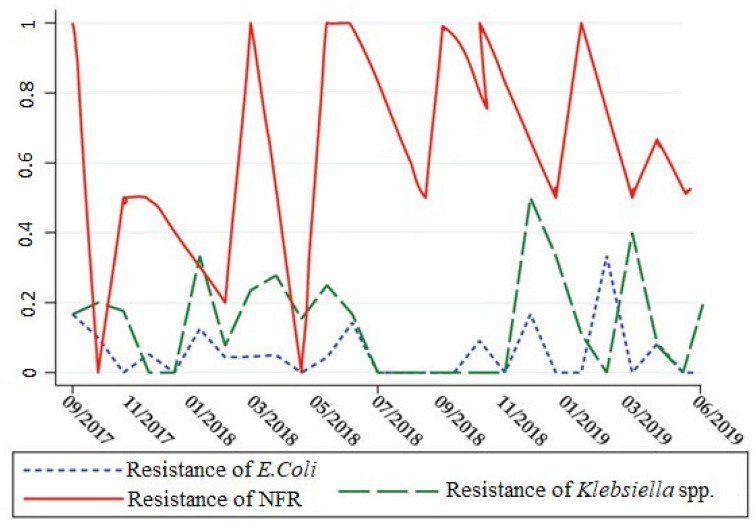
Monthly evolution of Meropenem resistance in *E. coli**, Klebsiella*, and Gram-negative non-fermenter rods (NFR).

**Figure 7 antibiotics-10-00868-f007:**
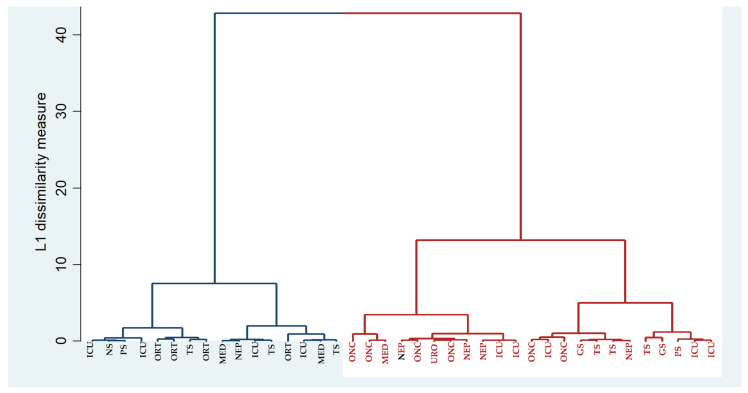
Clustering analys1is of the isolated *Klebsiella* strain using hospital ward as a label. ICU: Intensive Care Unit; NS: Neurosurgery; PS: Plastic Surgery; ORT: Orthopaedics; TS: Thoracic Surgery; MED: Medical wards; NEP: Nephrology; ONC: Oncology; URO: Urology; GS: General Surgery. Blue lines: low resistance cluster, red lines: high resistance cluster.

**Figure 8 antibiotics-10-00868-f008:**
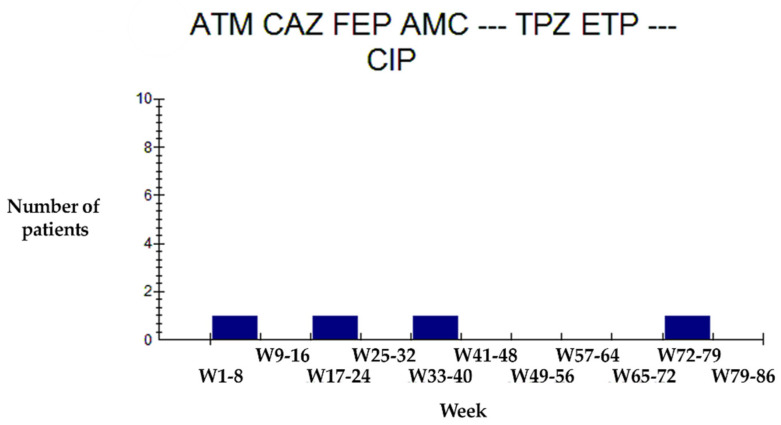
Evolution of the *Klebsiella* phenotype ATM-CAZ-FEP-AMC-TZP-ETP-CIP. Aztreonam (ATM) Ceftazidime (CAZ), Cefepime (FEP), Amoxicillin/Clavulanate (AMC) TPZ: Piperacillin/Tazobactam, Ertapenem (ETP) and Ciprofloxacin (CIP).

**Table 1 antibiotics-10-00868-t001:** The prevalence of *Klebsiella* strains in the various clinical specimens.

	No. Samples	No. Strains	Percent
Joint fluid	8	0	0.00%
Bile	5	0	0.00%
Blood	130	21	16.15%
Catheter	31	4	12.90%
Pleural fluid	19	0	0.00%
Cerebrospinal fluid	6	0	0.00%
Purulent secretion	276	34	12.32%
Sputum	128	35	27.34%
Urine	1294	77	5.95%
Tracheal aspirate	290	61	21.03%
Pharyngeal swab	72	4	5.56%
Nasal swab	38	5	13.16%
Puncture liquid	120	3	2.50%
Conjunctival secretion	32	4	12.50%
Ear discharges	7	3	42.86%
total	2456	251	10.22%

**Table 2 antibiotics-10-00868-t002:** Correlations between monthly ciprofloxacin resistance of *Staphylococcus aureus, E. coli, Klebsiella* and NFR.

	*Staphylococcus* *aureus*	*E. coli*	*Klebsiella*	NFR
***Staphylococcus aureus***	1			
***E. coli***	0.3472 *p* = 0.1336	1		
***Klebsiella***	0.4850 * *p* = 0.0302	0.2924 *p* = 0.2109	1	
**NFR**	−0.1583 *p* = 0.5051	0.2769 *p* = 0.2372	0.1359 *p* = 0.5679	1

* *p* < 0.05.

**Table 3 antibiotics-10-00868-t003:** Correlation between values of the monthly resistance index of *E. coli**, Klebsiella, Pseudomonas aeruginosa*, non-fermenter Gram-negative rods (NFR) and *Staphylococcus aureus*.

Species	*E. coli*	*Klebsiella*	*Ps. aeruginosa*	NFR	*Staphylococcus* *aureus*
*E. coli*	r = 1.0000				
*Klebsiella*	r = 0.5107 * *p* = 0.0214	r = 1.0000			
*Pseudomonas aeruginosa,*	r = 0.3184 *p* = 0.1841	r = 0.2755 *p* = 0.2536	r = 1.0000		
NFR	r = 0.1906 *p* = 0.4209	r = 0.1661 *p* = 0.4839	r = 0.3599 *p* = 0.1302	r = 1.0000	
*Staphylococcus aureus*	r = 0.1972 *p* = 0.4046	r = 0.3500 *p* = 0.1303	r = −0.4089 * *p* = 0.0422		r = 1.0000

* *p* < 0.05.

**Table 4 antibiotics-10-00868-t004:** Medium annualized antibiotic consumption in the hospital wards.

	No. Patients	AK (g)	AMC (g)	CRO (g)	CZO (g)	FEP (g)	CIP (g)	CAZ (g)	SXT (g)	IPM (g)	MEM (g)	TPZ (g)	TGC (g)
DDD/100 patient-days (WHO)		1	3	2	3	1.8	0.5	0.24	1	2	2	14	0.1
ICU	1646	1.01	0.9	12	3.5	1.27	0.16	0.11	0.35	1.83	6.5	23	0.04
Medical wards	1521	0.01	1.54	11.1	0.85	0	0.07	0.55	0	1.16	5.15	5.01	0.08
Nephrology	1003	0.08	0.7	16	1.49	1.25	0.03	0.08	0.36	2.83	7.4	12.29	0.02
Neurology	436	0.18	0.07	15	0	0.06	0.14	0.02	1.55	0	0.55	1.16	0
Oncology	804	0	0.95	9	0.44	0.29	0	0.04	0.07	0.32	2.47	0	0
Cardiology	219	0.09	12.9	16.17	0.13	0.3	0.49	0.4	0.25	2.45	2.54	1.24	0
Pediatrics	199	0.38	0.43	13.3	2.29	0	0.08	1.45	0	10.14	1.01	1.75	0
Pediatric surgery	25	0	0	0	0	0	0	0	0	0	0	0	0
Neurosurgery	1768	0.49	0.84	26	1.6	0.59	0.03	0.2	0	0	12.4	6.7	0
Ophthalmology	1696	0	0.15	15.8	0.04	0	0	0.11	0	0.05	0.21	0.77	0
Orthopaedics	2824	0	0.01	15.9	0.008	0.08	0.47	0	0	0	0.27	0	0
Urology	1927	0.98	0.06	18.8	0	0.03	0.02	0.58	0.06	3.63	1.17	0	0.003
General surgery	1329	0.6	0	17.6	6.32	0.009	0.12	0.12	1.7	1.44	3.61	1.49	0.003
Thoracic surgery	1131	0.52	0.12	19.3	0	2.28	0.26	0.55	0.08	0.25	4.22	9.75	0
Plastic surgery	34	0.39	0	15.5	0	0	1.3	0.33	0.07	0.35	0.6	1.86	0
Oral maxillofacial surgery	287	0	0	0	0	0	0	0.44	0	0	0	3.18	0

AK: Amikacin; AMC: Amoxicillin/Clavulanate; CAZ: Cephtazidime; CRO: Ceftriaxiaxone; CIP: Ciprofloxacin; CZO: Cephazolin; FEP: Cephepime; MEM: Meropenem; SXT: Sulphametoxazole/Trimethoprim; TGC: Tigecycline; TPZ: Piperacillin/Tazobactam.

**Table 5 antibiotics-10-00868-t005:** Linear regression analysis of correlations between antibiotic resistance in *Klebsiella* and antibiotic consumption.

Consumption of Antibiotics	Standardized Coefficient	Std. Err.	t	*p*	95% Confidence Interval
Ciprofloxacin	−0.0021	0.0008975	−2.34	0.101	[−0.0049579, 0.0007546]
Tigecycline	0.0118279	0.0032538	3.64	0.036 *	[0.001473, 0.0221829]
Piperacillin/ Tazobactam	−0.0027939	0.0008836	−3.16	0.050 *	[−0.0056061, 0.0000182]
Meropenem	0.003232	0.000955	3.38	0.043 *	[0.0001927, 0.0062714]
_constant	−0.3240754	0.1989285	−1.63	0.202	[−0.9571546, 0.3090037]

* *p* < 0.05.

**Table 6 antibiotics-10-00868-t006:** Time-series regression analysis of correlations between meropenem and Ciprofloxacin resistance in *Klebsiella* and antibiotic consumption.

Outcome Used	R Sq.	Predictors in the Model	Coefficient	95% Confidence Interval	*p*
Resistance to Meropenem	0.627	Consumption of Imipenem 1 month before	0.0243	[−0.0043, 0.0529]	0.078
		Consumption of Meropenem 1 month before	−0.0016	[−0.0057, 0.0025]	0.341
Resistance to Meropenem	0.753	Consumption of Imipenem 2 months before	0.0075	[0.0011, 0.0130]	0.032 *
		Consumption of Meropenem 2 months before	−0.0042	[−0.0077, 0.0081]	0.027
Resistance to Ciprofloxacin	0.753	Consumption of Imipenem 2 months before	0.0010	[0.0007, 0.0014]	<0.001 *
		Consumption of Meropenem 2 months before	−0.0002	[−0.0001, 0.0010]	0.094
		Consumption of Ciprofloxacin 2 months before	0.0004	[0.0001, 0.0007]	0.0047 *
Resistance to Ciprofloxacin	0.537	Consumption of Imipenem 3 months before	0.0064	[0.0004, 0.0009]	<0.001
		Consumption of Meropenem 3 months before	0.0001	[−0.0002, 0.0004]	0.571
		Consumption of Ciprofloxacin 3 months before	−0.0001	[−0.0004, 0.0001]	0.330

* *p* < 0.05.

## Data Availability

The data presented in this study are available on request from the corresponding author.

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
