# Peer review of "Prevalence and Antimicrobial Resistance of Klebsiella Strains Isolated from a County Hospital in Romania"

_antibiotics, 2021, doi:10.3390/antibiotics10070868_

Round 1

Reviewer 1 Report

Considering that one of the most important development of the last two decades in the field of antibiotic resistance is the emergence and spread of multiple antibiotic resistance bacteria in health-care facilities and especially in intensive care units that can be considered "factories" for creating, disseminating, and amplifying resistance to antibiotics, the subject and results of the article is very interesting and the paper deserves publication on Antibiotics.

However, some issues should be considered for manuscript improvement before publication:

  1. In the "Material and Methods" section it is necessary to indicate which antibiotics and which concentrations were used to perform the disk diffusion Kirby-Bauer method;
  2. The results reported in Figure 1 are not clear. Describe in detail the data shown in Figure 1 according to the percentages indicated on the abscissas and ordinates;
  3. Figure 6 describes monthly evolution of meropenem resistance in Escherichia coli, Klebsiella and Gram-negative non-fermenter rods. The results shown are up to 2018. It would be more correct to report the data obtained up to June 2019;
  4. In the "Discussion" section, at lines 348 and 349, the reported prevalence percentages of Klebsiella strains in ICU department and Thoracic Surgery and Orthopedics do not correspond to those shown in "Results" section at lines 188 and 189;
  5. The "References" section must be revised: some references are not appropriate (for example references 5 and 33), others not exhaustive (for example references 2 and 3);
  6. Minor errors in the text: i. indicate the names of the bacteria in italics; ii. after having reported for the first time the meaning of an acronym, do not specify it anymore in the text (for example Minimum Inhibitory Concentration (MIC), Intensive Care Unit (ICU), extended-spectrum beta-lactamase(ESBL); iii. Line 123 antiotheraphy correct with antibiotherapy; iv. Line 324 Ciprofloxacine correct with Ciprofloxacin; v. Line 344 stains correct with strains.

Author Response

Response to reviewers:

Author’s response: Thank you for your comments. It is our pleasure to have the opportunity to improve our manuscript. Point-to-point responses are given bellow.

1. Reviewer 1: In the "Material and Methods" section it is necessary to indicate which antibiotics and which concentrations were used to perform the disk diffusion KirbyBauer method.

Author’s response: Thank you for your observations. We added the required information in the text.

2. Reviewer 1: The results reported in Figure 1 are not clear. Describe in detail the data shown in Figure 1 according to the percentages indicated on the abscissas and ordinates;

Author’s response: We modified Figure 1 as suggested to be a simple bar graph instead of a Pareto histogram that is hard to understand, and added more explanations in the text.

3. Reviewer 1: Figure 6 describes monthly evolution of meropenem resistance in
Escherichia coli, Klebsiella and Gram-negative non-fermenter rods. The results shown are up to 2018. It would be more correct to report the data obtained up to June 2019;

Author’s response: Thank you. We made a mistake when we wrote the labels for abscissa, and we corrected it.

4. Reviewer 1: In the "Discussion" section, at lines 348 and 349, the reported prevalence percentages of Klebsiella strains in ICU department and Thoracic Surgery and Orthopedics do not correspond to those shown in "Results" section at lines 188 and 189;

Author’s response: We corrected the percentages in the „Discussion” section to match those in the „Results” section.

5. Reviewer 1: The "References" section must be revised: some references are not
appropriate (for example references 5 and 33), others not exhaustive (for example references 2 and 3);

Author’s response: We revised the reference list. We kept the reference 33 as an article showing the use on the strains’ clustering method based on MAR that can substitute the clustering performed by molecular methods as RFLP.

6. Reviewer 1: Minor errors in the text: i. indicate the names of the bacteria in italics; ii. after having reported for the first time the meaning of an acronym, do not specify it anymore in the text (for example Minimum Inhibitory Concentration (MIC), Intensive Care Unit (ICU), extended-spectrum beta-lactamase(ESBL); iii. Line 123 antiotheraphy correct with antibiotherapy; iv. Line 324 Ciprofloxacine correct with Ciprofloxacin; v. Line 344 stains correct with strains.

Author’s response: We corrected the errors in the text. We also modified Figure 4 in which also the antibiotic names were misspelled.

Additional comments from authors:
We performed a thorough English language revision and corrected the typos.

Reviewer 2 Report

In the world today, increased antimicrobial resistance (AMR) is a serious problem. Multidrug-resistant strains (MDR) emerge among bacteria, which poses a great threat to the health and life of patients. Kebsiella pneumoniae is one of the species belonging to the Enterobacteriaceae family, responsible for nosocomial and environmental infections. Research indicates that K. pneumoniae develops antibiotic resistance more easily than most bacteria by producing enzymes such as extended spectrum β-lactamase (ESBL). The species also plays a role in the transfer of antibiotic resistance genes from bacteria in the environment to bacteria of clinical importance.

The studies focus on the analysis of the prevalence of Klebsiella pneumoniae in various departments of the County Clinical Emergency Hospital of Craiova (SCJUC), Romania, and the monitoring of its resistance to antibiotics.

The language in the manuscript should be revised, as  there are certain errors (e.g. Figure 2, Figure 5, section 3.2 – timely= happening at the best possible moment. It should be changed; p. 7 l.221-222 “..due to hospital circulation of extended-spectrum beta-lactamase-producing (ESBL) strains in the hospital” – “"in the hospital"  seems redundant”).

It should be noted that there are a number of errors in the work. For example:

- in most cases, throughout the work, the Latin names of bacteria are not italicized

- in the name, e.g. Klebsiella spp., the first part is written in italics, the second without italics

- the full species name is given when we write it for the first time. Later we can use the abbreviated name

- Klebsiella pneumoniae (K. pneumoniae) the abbreviation need not be used - the name in brackets can be removed

Materials and methods

- ,,… ..the period 1.09.2017-31.06.2019… ”- June has 30 days

- names of hospital wards, sometimes with a capital letter, sometimes with a small letter (repeated throughout the text)

- ,,… Minimum Inhibitory Concentration the MIC (MIC Minimum Inhibitory Concentration)…. ”- repeats, remove brackets and “the” before the abbreviation

-once it says "... WHONET .." then "... Whonet .."

Results

- ,, .. was found in a percentage of 10.22%, .. ”- too many percent

- Thoracic surgery (23.22%), Urology (22.58%), Medical ward (21.32%), General surgery (20.98%), where it is a causative agent of decubitus pneumonia, most of these patients being bet. - => the phrase “most of these patients being bet” is unclear

-In Abstract, it says - “…. that ranged between 5.25% and 19.49% in June 2018, while in medical wards the variation was much wider, between 5.15% and 36.36% in April 2018. " In turn, in the Results - ,, ... that ranged between 5.25% and 19.49% in July 2018, while in medical wards, the variation ranged between 5.15% and 17.36% in April 2018 "- which month is meant: June or July and why are the different values given: 36.36% in April, and then 17.36% in April. Looking at Fig. 2, it should probably be 17.36%.

- what does No. On the ordinate in Fig. 3 mean?

- ,,… Klebsiella amoxicillin/clavulanate (85.31%), ceftriaxone (63.51%), ceftazidime (76.62%), cefepime (66.81%), aztreonam (43.67%). Note the resistance to carbapenems due to the circulation of plasmids encoding carbapenemase-producing genes. Thus, the resistance to Ertapenem versus Klebsiella was 27.78%. Resistance to quinolones (ciprofloxacin) in Klebsiella strains was 30.57% "

Some of the values ​​given in the text do not agree with the data from Figure 4

-page 8 where do the data for Staphylococcus aureus, Escherichia coli and NFR come from? there are no references

-Figure 6 - where do the data for Escherichia coli and NFR come from

-same as above for Table 3. Where do the data come from? Additionally, the value in the text does not agree with that given in the table. Text - ,, Pseudomonas aeruginosa and Staphylococcus aureus (r = -0.4089, p = 0.0822) "table - (r = -0.4089, p = 0.0422)

-page 10 - information in brackets - "(Error! Reference source not found.)" ?!

-Table 4 - the table gives the abbreviation of TPZ in the explanations of the TZP. There are also no spaces before FEP

- names of antibiotics, sometimes with a capital letter, sometimes with a lowercase letter

-page 11 in the description above the table, there is “…. Moxifloxacine (0.70%) ”- should be Moxifloxacin. In addition, Ciprofloxacin is listed in place of Moxifloxacin in Table 5

-page 13 ,, The dissemination of hospital stains… ”- should be written - strains

-page 14 - ,,… in prevalence rates of Klebsiella pneumoniae and others. "- what does others mean here?

Literature

Please, unify the text - all the words in the title should be written with uppercase or lowercase letter

Author Response

Response to reviewers:

Author’s response: Thank you for your comments. It is our pleasure to have the opportunity to improve our manuscript. Point-to-point responses are given bellow.

Reviewer 2: The language in the manuscript should be revised, as there are certain errors (e.g. Figure 2, Figure 5, section 3.2 – timely= happening at the best possible moment. It should be changed; p. 7 l.221-222 “..due to hospital circulation of extended-spectrum beta-lactamase producing (ESBL) strains in the hospital” – “"in the hospital" seems redundant”).

Author’s response: Thank you for your observation. We deleted the word ‘timely’.

1. Reviewer 2: It should be noted that there are a number of errors in the work. For example:
- in most cases, throughout the work, the Latin names of bacteria are not italicized

Author’s response: Thank you for your observation. We italicized the Latin names.

2. Reviewer 2:- in the name, e.g. Klebsiella spp., the first part is written in italics, the second without italics

Author’s response: Thank you for your observation. We italicized the Latin names.

3. Reviewer 2:- the full species name is given when we write it for the first time. Later we can use the abbreviated name.

Author’s response: Thank you for your observation. At your suggestion, we used the abbreviations.

4. Reviewer 2:-Klebsiella pneumoniae (K. pneumoniae) the abbreviation need not be used -the name in brackets can be removed.

Author’s response: Thank you for your observation. At your suggestion, we used the abbreviation.

5. Reviewer 2: Materials and methods
- ,,… ..the period 1.09.2017-31.06.2019… ”- June has 30 days

Author’s response: Thank you for your observation. We made the correction.

6. Reviewer 2: names of hospital wards, sometimes with a capital letter, sometimes with a small letter (repeated throughout the text)

Author’s response: Thank you for your observation. We made the correction, all hospital wards are now with capital letter.

7. Reviewer 2: ,,… Minimum Inhibitory Concentration the MIC (MIC Minimum Inhibitory Concentration)…. ”- repeats, remove brackets and “the” before the abbreviation

Author’s response: Thank you for your observation. We made the correction.

8. Reviewer 2: -once it says "... WHONET .." then "... Whonet .."

Author’s response: Thank you for your observation. We corrected the mistake.

9. Reviewer 2: Results- ,, .. was found in a percentage of 10.22%, .. ”- too many percent

Author’s response: Thank you for your observation. We calculated this percent as 251 strains from all 2456 samples. We rephrased the text.

10. Reviewer 2: Thoracic surgery (23.22%), Urology (22.58%), Medical ward (21.32%), General surgery (20.98%), where it is a causative agent of decubitus pneumonia, most of these patients being bet. - => the phrase “most of these patients being bet” is unclear.

Author’s response: Thank you for your observation. We removed this unclear sentence.

11. Reviewer 2: -In Abstract, it says - “…. that ranged between 5.25% and 19.49% in June 2018, while in medical wards the variation was much wider, between 5.15% and 36.36% in April 2018. " In turn, in the Results - ,, ... that ranged between 5.25% and 19.49% in July 2018, while in medical wards, the variation ranged between 5.15% and 17.36% in April 2018 "- which month is meant: June or July and why are the different values given: 36.36% in April, and then 17.36% in April. Looking at Fig. 2, it should probably be 17.36%.

Author’s response: Thank you for your observation. We corrected and added June in Results. Also we corrected the value at 17.36% in Abstract.

12. Reviewer 2:- what does No. On the ordinate in Fig. 3 mean?

Author’s response: Thank you for your observation. We added the necessary information.

13. Reviewer 2:- ,,… Klebsiella amoxicillin/clavulanate (85.31%), ceftriaxone (63.51%), ceftazidime (76.62%), cefepime (66.81%), aztreonam (43.67%). Note the resistance to carbapenems due to the circulation of plasmids encoding carbapenemase-producing genes. Thus, the resistance to Ertapenem versus Klebsiella was 27.78%. Resistance to quinolones (ciprofloxacin) in Klebsiella strains was 30.57% " Some of the values given in the text do not agree with the data from Figure 4.

Author’s response: Thank you for your observation. We correlated the values in the text with those from Figure 4.

14. Reviewer 2: -page 8 where do the data for Staphylococcus aureus, Escherichia coli and NFR come from? there are no references.
-Figure 6 - where do the data for Escherichia coli and NFR come from.

Author’s response: Thank you for your observation. From all samples analyzed (2456), we isolated 251 strains of Klebsiella spp. We also isolated 1252 strains of other bacteria, like: Staphylococcus aureus, Escherichia coli and NFR. We added these information in the Results section.

15. Reviewer 2: -same as above for Table 3. Where do the data come from? Additionally, the value in the text does not agree with that given in the table. Text - ,,Pseudomonas aeruginosa and Staphylococcus aureus (r = -0.4089, p = 0.0822) "table - (r = -0.4089, p = 0.0422)

Author’s response: Thank you for your observation. We corrected the p-values.

16. Reviewer 2: -page 10 - information in brackets - "(Error! Reference source not found.)" ?!

Author’s response: Thank you for your observation. We corrected the errors

17. Reviewer 2: -Table 4 - the table gives the abbreviation of TPZ??? in the explanations of the TZP. There are also no spaces before FEP

Author’s response: Thank you for your observation. We corrected these mistakes.

18. Reviewer 2:- names of antibiotics, sometimes with a capital letter, sometimes with a lowercase letter

Author’s response: Thank you for your observation. We made the correction, all name of antibiotics are now with capital letter

19. Reviewer 2:-page 11 in the description above the table, there is “…. Moxifloxacine (0.70%) ”- should be Moxifloxacin. In addition, Ciprofloxacin is listed in place of Moxifloxacin in Table 5

Author’s response: Thank you for your observation. We made the required modification in Table 5.

20. Reviewer 2:-page 13 ,, The dissemination of hospital stains… ”- should be written – strains

Author’s response: Thank you for your observation. We corrected this mistake.

21. Reviewer 2: -page 14 - ,,… in prevalence rates of Klebsiella pneumoniae and others. "- what does others mean here?

Author’s response: Thank you for your observation. We reformulated this phrase. (“others” refers to other studies)

22. Reviewer 2: Literature
Please, unify the text - all the words in the title should be written with uppercase or lowercase letter

Author’s response: Thank you for your observation. We made this correction.

Additional comments from authors:
We performed a thorough English language revision and corrected the typos.

Reviewer 3 Report

1, the content of Introduction was to many un-relevant words, please  simplify it.

2, Klebsiella spp includes many kind of species, which incur different infections and their resistance is also difference. it is better to investigate only one major  specie such K. pneumoniae. 

3, most of the sample was respiratory samples, it is difficulty to differentiate colonization, contamination and infection. the authors should clearly identify all the strains were infectious agents.

4, what is the orange curve in the figure 1?

5, the legend of figure 3 was not clear. it was very difficulty to read.

6, Table 3 was a non-sense  data.

7, figure 7 can not give any valuable information about relationship of resistance.

Author Response

Response to reviewers:
Author’s response: Thank you for your comments. It is our pleasure to have the opportunity to improve ourmanuscript. Point-to-point responses are given bellow.

1. Reviewer 3: the content of Introduction was to many un-relevant words, please simplify it.

Author’s response: Thank you for the observation. We revised the introduction.

2. Reviewer 3: Klebsiella spp includes many kind of species, which incur different infections and their resistance is also difference. It is better to investigate only one major specie such K. pneumoniae.

Author’s response: Thank you for the observation. We identified bacteria by classical microbiological methods that cannot distinguish between species of Klebsiella. We only identified the multiresistant strains to the species level using the automated analyzer. So, we considered that if we analyze only K. pneumoniae, then we would analyze only the multiresistant strains, which will induce a bias and make impossible the analysis of the distribution of resistance
level, the relationship with antibiotic consumption and can severely limit the analysis of filiation of strains based on resistance index. These analyses all require to investigate all Klebsiella strains identified from the respective ward.

3. Reviewer 3: most of the sample was respiratory samples, it is difficulty to differentiate colonization, contamination and infection. The authors should clearly identify all the strains were infectious agents.

Author’s response: Thank you for the observation. The study was focused on analyzing the distribution of resistance inside and between the wards, on investigating the relationship with consumption of antibiotics and the analysis of strain relatedness. These analyses require to investigate all the strains, not only the ones producing infection, in order to obtain meaningful results which can be used to adjust the infection control measures.

4. Reviewer 3: what is the orange curve in the figure 1?

Author’s response: thank you for your observations. We eliminate the orange curve and explain better figure 1

5. Reviewer 3: the legend of figure 3 was not clear. it was very difficulty to read.

Author’s response: Thank you for the observation. We clarified the legend of Figure 3.

6. Reviewer 3: Table 3 was a non-sense data.
Author’s response: Thank you for the observation. In table 3 we showed a matrix of correlations between antibiotic resistance in different bacteria. It is needed to examine all possible combinational correlations in order to identify a significant link between resistances. This is important because association between resistance of different bacteria can mean a common cause (eg dissemination of resistance genes or consumption of antibiotics).

7. Reviewer 3: figure 7 can not give any valuable information about relationship of resistance.

Author’s response: Thank you for the observation. It can be observed from the figure that strains from the same ward have a closer relationship of antibiotic resistance that strains from different wards. This can help identify the problematic wards in which transmission of resistant strains is frequent in order to take infection control measures.

Additional comments from authors:
We performed a thorough English language revision and corrected the typos.

Reviewer 4 Report

The work presented by Ghenea et al. studies the prevalence and antimicrobial resistance of Klebsiella strains isolated from a hospital, which is very interesting. To achieve this goal, appropriate procedures and analyses have been employed. I recommend accepting the manuscript after the following revisions.

Specific comments

Line 88 and throughout the text: Species and genus names should go in italics.

Lines 137-138: Please, provide the Minimum Inhibitory Concentration (MIC).

Line 143: "Organization"

Line 193: Please, explain here the meaning of the blue bars as well as the red line. Also provide a title and explain the meaning of the percentages shown in the two Y axis.

Line 205, Figure 2: Please, provide a title to the Y axis

Line 220, Figure 3: Please, explain the meaning of the columns and the green line. Some of the figures lack the values in the X axis, please, provide them.

Line 241, Figure 5: Please, provide a title to the Y -axis.

Lines 250-251, 274,275: Please, provide the appropriate reference here.

Line 268: The value p=0.0822 does not correspond with the one present in Table 3. Please, clarify this point and assign an asterisk in the table if its value is correct and significant.

Line 281: Please, replace "one can encounter" by "it's possible to find" or something less colloquial.

Line 291: Moxifloxacine is not shown in Table 5, please, clarify it.

Line 293: In Table 5, what is the meaning of "_constant"? Please, clarify this point."

Line 314: "that the ICU strains were aggregated"

Lines 314-315: This sentence is too long, please split it up into two sentences.

Line 311, Figure 7: Please, provide a color code to distinguish the low and high resistance indexes in this figure.

Lines 322-325: "The following antibiotics with common resistances were used to define resistance profiles: Aztreonam (ATM)... "

Line 325: "We analysed the next in our strains the frequency of...". Please, check the meaning of this sentence.

Line 233: Please, indicate what is represented in the X-axis.

Line 384: Higher what? Please, clarify it.

Lines 393-394: Please, give a reference on the transmission of plasmids encoding resistance genes.

Line 414: Please, replace "the former" by "the latter".

Line 414: Is there a reference pointing out this interesting feature of Klebsiella?

Line 455: Also, simply because of the presence of uncultured members which may be reluctant to be grown under laboratory conditions.

Author Response

Response to reviewers:
Author’s response:
Thank you for your comments. It is our pleasure to have the opportunity to improve our manuscript. Point-to-point responses are given bellow.

1. Reviewer 4: Line 88 and throughout the text: Species and genus names should go in italics.
Author’s response: Thank you for your observation. We put them in italics.

2. Reviewer 4: Lines 137-138: Please, provide the Minimum Inhibitory Concentration (MIC).

Author’s response: Thank you for your observation. We provided MIC.

3. Reviewer 4: Line 143: "Organization"

Author’s response: Thank you for your observation. We corrected the error.

4. Reviewer 4: Line 193: Please, explain here the meaning of the blue bars as well as the red line. Also provide a title and explain the meaning of the percentages shown in the two Y axis.

Author’s response: Thank you for your observation. We eliminate the red line and better explain figure 1 bars meaning in the legend. The bars heights are the percentages of prevalence of Klebsiella strains in different hospital wards.

5. Reviewer 4: Line 205, Figure 2: Please, provide a title to the Y axis

Author’s response: Thank you for your observation. We provided the title of the Y axis as suggested.

6. Reviewer 4: Line 220, Figure 3: Please, explain the meaning of the columns and the green line. Some of the figures lack the values in the X axis, please, provide them.

Author’s response: Thank you for your observation. The green line is a mathematical smoothing of all the bars in the histogram, in order to provide a visual cue of the shape of the frequency distribution (eg. Like in the Gauss curve). We provided the title of the X and Y axis as suggested.

7. Reviewer 4: Line 241, Figure 5: Please, provide a title to the Y-axis.

Author’s response: Thank you for your observation. We provided the title of the Y axis as suggested.

8. Reviewer 4: Lines 250-251, 274,275: Please, provide the appropriate reference here.
Author’s response: Thank you for your observation. At these lines is only our results presented and they don’t require a reference. The appropriate references were in the Discussion section when we discussed these results.

9. Reviewer 4: Line 268: The value p=0.0822 does not correspond with the one present in Table 3. Please, clarify this point and assign an asterisk in the table if its value is correct and significant.

Author’s response: Thank you for your observation. We corrected the P value in the text and added an asterisk in the table.

10. Reviewer 4: Line 281: Please, replace "one can encounter" by "it's possible to find" or something less colloquial.

Author’s response: Thank you for your observation. We replaced the text.

11. Reviewer 4: Line 291: Moxifloxacine is not shown in Table 5, please, clarify it.

Author’s response: We deleted Moxifloxacine from the text, as it was added by error (correct Ciprofloxacine, but its effect is non-significant).

12. Reviewer 4: Line 293: In Table 5, what is the meaning of "_constant"? Please, clarify this point."

Author’s response: Thank you for your observation. The constant term means the predicted resistance by the model that is independent of the antibiotic consumption (if all the coefficients were 0). We added this in the text.

13. Reviewer 4: Line 314: "that the ICU strains were aggregated"

Author’s response: Thank you for your observation. We replace “aggregated” by divided, as the ICU strains were split in 3 clusters according to the level of resistance.

14. Reviewer 4: Lines 314-315: This sentence is too long, please split it up into two sentences.

Author’s response: Thank you for your observation. We split the phrase as suggested.

15. Reviewer 4: Line 311, Figure 7: Please, provide a color code to distinguish the low and high resistance indexes in this figure.

Author’s response: Thank you for your observation. We modified the figure.

16. Reviewer 4: Lines 322-325: "The following antibiotics with common resistances were used to define resistance profiles: Aztreonam (ATM)... "

Author’s response: Thank you for your observation. We modified the text to be more concise.

17. Reviewer 4: Line 325: "We analysed the next in our strains the frequency of...". Please,check the meaning of this sentence.

Author’s response: Thank you for your observation. We modified the text to be more concise.

18. Reviewer 4: Line 233: Please, indicate what is represented in the X-axis.

Author’s response: Thank you for your observation. We added the title for the X axis.

19. Reviewer 4: Line 384: Higher what? Please, clarify it.

Author’s response: Thank you for your observation. We modified the text.

20. Reviewer 4: Lines 393-394: Please, give a reference on the transmission of plasmids encoding resistance genes.

Author’s response: Thank you for your observation. We added the reference.

21. Reviewer 4: Line 414: Please, replace "the former" by "the latter".

Author’s response: Thank you for your observation/ We replaced the text as suggested.

22. Reviewer 4: Line 414: Is there a reference pointing out this interesting feature
of Klebsiella?

Author’s response: Thank you for your observation. We added references as suggested.

23. Reviewer 4: Line 455: Also, simply because of the presence of uncultured members which may be reluctant to be grown under laboratory conditions.

Author’s response: Thank you for your observation. At your suggestion, we reformulated the Limitations section.

Additional comments from authors:
We performed a thorough English language revision and corrected the typos

Reviewer 5 Report

Thank you for this interesting article. Figure 1 needs a description of the x and y-axis more clearly.

I did not understand why multiple times in the article some analysis are marked with "(Error! Reference source not found.)"

The conclusion should be definite/to the point without the use of extra words. Like just mention the finding, no need to explain/elaborate in the conclusion. 

Antibiotic resistance of Klebsiella is not a new problem and authors have also mentioned the mechanism of antibiotic resistance in general/in the global setting. But authors should mention any unique point about this study separately (maybe a subheading) in the discussion to make the manuscript more interesting for the readers. 

Author Response

Response to reviewers:
Author’s response:
Thank you for your comments. It is our pleasure to have the opportunity to improve our manuscript. Point-to-point responses are given bellow.

1. Reviewer 5: Thank you for this interesting article. Figure 1 needs a description of the x and y-axis more clearly.

Author’s response: We modified Figure 1 as suggested to be a simple bar graph instead of a Pareto histogram that is hard to understand, and added more explanations in the text.

2. Reviewer 5: I did not understand why multiple times in the article some analysis are marked with "(Error! Reference source not found.)"

Author’s response: Thank you for your observation. Those were automated references to numbered figures that didn’t compile on your version of Word. The issue was resolved.

3. Reviewer 5: The conclusion should be definite/to the point without the use of extra words. Like just mention the finding, no need to explain/elaborate in the conclusion.

Author’s response: Thank you for your observation. We revised the conclusions.

4. Reviewer 5: Antibiotic resistance of Klebsiella is not a new problem and authors have also mentioned the mechanism of antibiotic resistance in general/in the global setting. But authors should mention any unique point about this study separately (maybe a subheading) in the discussion to make the manuscript more interesting for the readers.

Author’s response: Thank you for your observation. We added subheadings in the Discussion section and detailed the time-series analysis of the link between antibiotic resistance and antibiotic consumption.

Additional comments from authors:
We performed a thorough English language revision and corrected the typos.

Round 2

Reviewer 2 Report

There are still some errors in the manuscript. I ask the authors to take into account my suggestions.  The errors in the text include, among others

-  the Latin names of bacteria should be italicized e.g. Fig. 2;  Table 2; Figure 6;

- the name, e.g. Klebsiella – should be  written in italics;  spp. - without italics

- Klebsiella pneumoniae (K. pneumoniae), Escherichia coli (E.coli) -  the abbreviation need not be used,  the name in the brackets can be removed. It makes no sense to repeat the same name,  e.g. page 2 lines 84, 85

-Table 1 - there is no information on the frequency of Klebsiella strains in purulent secretion

-Table 2; Table  5 – The antibiotic names should be written with uppercase letter

-Table 5 - there should be a slash between Piperacillin and Tazobactam

-References – position 15 - ,, Clin. Infect. Dis. an Off. Publ. Infect. Dis. Soc. Am.”  - the correct abbreviation is - Clin Infect Dis Off Publ Infect Dis Soc Am.

Position 32 - Please, unify text - the words in the title should be written with lowercase letter

the Latin names of bacteria should be italicized

Author Response

Response to reviewers:

Thank you for your comments. Point-to-point responses are given bellow.

  1. Reviewer 2: - the Latin names of bacteria should be italicized e.g. Fig. 2;  Table 2; Figure 6;

Author’s response: Thank you for your observation. We italicized the Latin names in Figure 2, Table 2 and Figure 6.

  1. Reviewer 2: - the name, e.g. Klebsiella – should be written in italics; spp. - without italics

Author’s response: Thank you for your observation. We made the correction in all manuscript.

  1. Reviewer 2: - Klebsiella pneumoniae (K. pneumoniae), Escherichia coli (E.coli) - the abbreviation need not be used, the name in the brackets can be removed. It makes no sense to repeat the same name, e.g. page 2 lines 84, 85

Author’s response: Thank you for your observation. We made the required correction in the text.

  1. Reviewer 2: -Table 1 - there is no information on the frequency of Klebsiella strains in purulent secretion

Author’s response:  Thank you for your observation. I apologize for the mistake of translation in Table 1, when I realized the statistics. Actually the wound secretion from Table 1 is purulent secretion. We made this correction in Table 1.

  1. Reviewer 2: -Table 2; Table 5 – The antibiotic names should be written with uppercase letter

Author’s response: Thank you for your observation. We made these corrections in Table 2 and Table 5.

  1. Reviewer 2: -Table 5 - there should be a slash between Piperacillin and Tazobactam

Author’s response: Thank you for your observation. We made this correction.

  1. Reviewer 2: -References – position 15 - ,, Clin. Infect. Dis. an Off. Publ. Infect. Dis. Soc. Am.” - the correct abbreviation is - Clin Infect Dis Off Publ Infect Dis Soc Am.

Author’s response: Thank you for your observation. We made this correction.

  1. Reviewer 2: Position 32 - Please, unify text - the words in the title should be written with lowercase letter

Author’s response: Thank you for your observation. We put the articles titles with lowercase letter.

  1. Reviewer 2: the Latin names of bacteria should be italicized

Author’s response: Thank you for your observation. We italicized all Latin names of bacteria.
